# Neuro Symbolic Graph Generative Modeling

## Abstract

We challenge the prevailing deep generative paradigm for graphs, exemplified by diffusion models, which is computationally intensive, lacks formal guarantees, and offers little user control. We introduce Neuro-Symbolic Graph Generative Modeling (NSGGM), which recasts graph generation as sequence modeling plus constraint satisfaction. NSGGM learns a vocabulary of subgraph tokens and uses an autoregressive sampler to propose token sequences, which an SMT solver then assembles into valid graphs by enforcing learned structural rules and user-defined constraints. This hybrid design avoids costly iterative refinement while providing correctness by construction and interpretable control. Across molecular and general graph benchmarks, NSGGM achieves state-of-the-art quality with fine-grained, user-steerable control, which current methods lack. Thus, NSGGM offers a practical path to trustworthy, targeted graph synthesis with broad applicability.

## 1 Introduction

Generative modeling of graph-structured data is a fundamental challenge with profound implications for scientific discovery and engineering (Brockschmidt et al., 2019). From designing novel molecules and materials (Liu et al., 2018a; Cao & Kipf, 2018) to discovering electronic circuits (Chang et al., 2024), the ability to generate valid and optimized graphs is a key driver of innovation. However, generating graphs remains difficult due to their non-sequential nature, sparsity, and the need to preserve structural fidelity. In recent years, deep generative models, particularly diffusion models (Vignac et al., 2023a; Xu et al., 2024; Limnios et al., 2024), have emerged as the state-of-the-art, demonstrating impressive capabilities in capturing complex data distributions.

Despite their success, diffusion-based methods face practical limits inherent to end-to-end neural design. They are *computationally expensive*, relying on slow, iterative refinement (often hundreds of steps; e.g., DiGress uses 500 (Vignac et al., 2023a)), inflating training and inference cost. They *lack formal guarantees*, so invalid structures (e.g., chemically nonsensical molecules) require post-hoc filtering. Their *black-box* nature hinders interpretability—problematic in domains like drug discovery where transparency is crucial (Amann et al., 2020; Bussmann et al., 2021). Relatedly, user control is *limited and weakly grounded*: the same opacity pushes steering into latent/score space via surrogates and soft penalties rather than explicit, verifiable graph constraints. Collectively, these issues restrict deployment where validity, interpretability, and controllability are essential.

To this end, we introduce **N**euro-**S**ymbolic **G**raph **G**enerative **M**odeling (NSGGM), a novel neural–symbolic framework reframing graph generation as a dual problem of sequence generation and constraint-satisfaction problem solving. Our approach begins by decomposing graphs into a vocabulary of building blocks (subgraph tokens) equipped with *interface information* specifying how they may connect to neighboring tokens, while deriving a set of symbolic assembly constraints over these interfaces. Thanks to constraints, a graph can be viewed as a sequence of tokens, analogous to words in natural language, enabling efficient sequential modeling while maintaining structural validity. A neural decoder is trained autoregressively to learn probability distribution over valid token sequences. At inference, the decoder proposes a complete sequence in a single forward pass, which serves as a blueprint. This blueprint is passed to a symbolic SMT solver (Barrett & Tinelli, 2018) that deterministically constructs the final graph by satisfying the derived constraints.

While we share the idea of assembling predefined substructures with prior fragment-based approaches (Jin et al., 2018b; Mercado et al., 2021a; Jin et al., 2020b), the key distinction is how assembly is performed. Those pipelines rely on heuristic search or handcrafted rules over a fixed library,

making global constraints hard to express or modify (limiting flexibility) and offering no formal guarantees—limitations shared by prevailing diffusion-based methods. NSGGM takes a neuro-symbolic approach that directly addresses these shortcomings by separating fast, learned planning from exact, verifiable assembly. Concretely, we emphasize two advantages: i) *Controllability*: we enable fine-grained, user-steerable control via interpretable constraints and explicit solver reasoning; users can guide generation with partial structural prompts (e.g., include a specific ring system or scaffold) and declarative constraints that require or forbid motifs, bound ring counts, or limit distances, all enforced exactly at assembly time. (ii) *Verifiability*: the solver provides formal correctness guarantees, so outputs are valid by construction and auditable rather than filtered post hoc. Our empirical results across molecular and graph benchmarks show that NSGGM achieves perfect validity and competitive quality compared to state-of-the-art methods, while offering greater interpretability, user control, and solver-backed verifiability. In particular, we demonstrate a clear advantage over existing methods in strong conditional controllability. Therefore, NSGGM points to a practical path for trustworthy, targeted graph synthesis with broad applicability, such as drug design. In summary, our key contributions are:

- We introduce NSGGM, a novel neuro–symbolic framework that reframes graph generation as a dual problem of sequence modeling and constraint-satisfaction.

- We introduce subgraph tokenization with typed interfaces and symbolic assembly constraints. A lightweight autoregressive decoder emits a single-pass *blueprint* that an SMT solver compiles into a graph with validity by construction and auditable guarantees.

- NSGGM offers fine-grained, user-steerable control along two complementary axes: (i) prompt-based completion from user-supplied partial structures, and (ii) constraint-driven synthesis to satisfy user-specified property or topology requirements.

- We demonstrate that NSGGM achieves state-of-the-art performance on molecular and general graph benchmarks, with an emphasis on conditional generation and flexible user control, substantiated by quantitative evaluations and case studies.

## 2 PRELIMINARIES

### 2.1 GRAPH GENERATION TASKS

The primary goal of graph generation is to learn the underlying distribution from a dataset of graphs $\mathcal{G} = \{G_1, \ldots, G_n\}$ in order to synthesize new, valid samples. We focus on graphs with categorical node and edge attributes, a common setup in molecular design (Vignac et al., 2023a). Formally, we define a graph as a tuple $G = (V, E, \mathbf{x}, \mathbf{e})$, where $V$ is the set of nodes, $E \subseteq V \times V$ is the set of edges, and $\mathbf{x} : V \to \mathcal{X}$ and $\mathbf{e} : E \to \mathcal{E}$ assign attributes from discrete label sets $\mathcal{X}$ and $\mathcal{E}$, respectively. The task is then to learn a parameterized model $P_\theta(G)$ that approximates the true data distribution and to use it for generating novel graphs $G' \sim P_\theta(G)$.

Prevailing approaches fall into two families: (i) diffusion models, which learn $P_\theta(G)$ by minimizing a divergence (typically KL) and generate graphs via computationally inefficient, iterative denoising (Yang et al., 2024); and (ii) fragment-based methods that assemble graphs from predefined substructures using heuristic search or hard-coded rules. Despite strong generative performance, both offer limited controllability and lack formal correctness guarantees under real-world design constraints.

### 2.2 CONSTRAINT SATISFACTION PROBLEM SOLVING AND SMT SOLVERS

The deterministic assembly stage of our framework is formulated as a Constraint Satisfaction Problem (CSP), necessitating a tool with formal guarantees. For this, we turn to Satisfiability Modulo Theories (SMT) solvers. While any off-the-shelf SMT solver could be employed, we choose Z3 (de Moura & Bjørner, 2008) as it is considered state-of-the-art in performance and reliability.

A theory $T$ in first-order logic defines a set of symbols (e.g., constants like 0, functions like $+$) and axioms that constrain their interpretation (e.g., the theory of linear integer arithmetic, $T_{\text{LIA}}$). An SMT solver's task is to determine if a given quantifier-free first-order formula $\phi$ is *T-satisfiable*—that is, if there exists a model that satisfies both the axioms of $T$ and the formula $\phi$. In this work, the formula $\phi$ will be the encoding of constraints used to assemble a valid graph from a proposed blueprint.

The problem is typically defined by a logical formula $\phi$ over a set of variables $Z = \{z_1, \ldots, z_m\}$, where each variable $v_i$ has a corresponding domain $D_i$. The formula is a conjunction of constraints

$$C = \{c_1, \ldots, c_n\}: \quad \phi := \bigwedge_{j=1}^{n} c_j(Z_j), \quad Z_j \subseteq Z. \tag{1}$$

A solution is a *model $M$*, which is an interpretation mapping each variable to a value in its domain, $M : Z \to D$, such that the formula is satisfied ($M \models \phi$). The solver returns one of two outcomes:

- SAT (Satisfiable): A valid model $M$ exists and is returned.
- UNSAT (Unsatisfiable): The solver proves that no such model exists.

The model $M$ returned by the solver on a SAT result provides the deterministic instructions for constructing a valid graph. We detail our specific constraint encoding $\phi$ in Section 3.2.

## 3 THE NSGGM FRAMEWORK

Unlike prevailing end-to-end deep generative models, our NSGGM framework treats graph generation as a compositional task, analogous to building with modular components. Our method first decomposes existing graphs into a vocabulary of fundamental pieces. A new, valid graph is then created by sampling from this vocabulary and assembling the pieces while adhering to symbolic assembly constraints. Figure 1 provides an overview of this neuro-symbolic framework.

### 3.1 GRAPH DECOMPOSITION AND VOCABULARY CONSTRUCTION

The foundation of our framework is transforming each graph $G$ from the training dataset $\mathcal{G}$ into a high-level compositional representation, or *blueprint*. This is achieved by a deterministic decomposition that partitions a graph into reusable, annotated subgraph tokens. The collection of all unique token types across $\mathcal{G}$ forms our vocabulary $\mathcal{V}$.

**Structural partitioning.** Our decomposition leverages the graph's cycle structure, akin to fragment-based methods (e.g., Jin et al., 2018b; Mercado et al., 2021a). For $G = (V, E)$, we compute a minimum cycle basis $\mathcal{C}(G) = \{c_1, \ldots, c_p\}$ with edge sets $E(c_j)$ and split edges into

$$E_{\mathcal{C}} = \bigcup_{j=1}^{p} E(c_j), \qquad E_A = E \setminus E_{\mathcal{C}}. \tag{2}$$

This induces primitives

$$\mathcal{P}(G) = \mathcal{C}(G) \cup \text{Components}(G_A), \quad G_A = (V, E_A), \tag{3}$$

which we enumerate as $\mathcal{P}(G) = \{P_i = (V_i, E_i)\}_{i=1}^{m}$. The primitives form an *overlapping cover* of $G$ (their nodes/edges union to $V$ and $E$). We make overlaps explicit via

$$V_{\text{sh}} = \bigcup_{i \neq j}(V_i \cap V_j), \qquad E_{\text{sh}} = \bigcup_{i \neq j}(E_i \cap E_j). \tag{4}$$

For the acyclic part, we refine components into recurring *tree motifs* (Appendix A). This design largely echoes Jin et al. (2018b), with one key difference: whereas Jin et al. (2018b) use single edges (plus rings) as acyclic clusters, we admit multi-edge *tree* motifs and equip them with explicit slot interfaces. Overlaps ($V_{\text{sh}}, E_{\text{sh}}$) are preserved and reconciled during assembly by slot matching; no explicit projection maps are required.

**Interface characterization.** To enable faithful reassembly, we define node and edge slot assignments, $\sigma_V$ and $\sigma_E$. A node $v \in V$ is *shared* if $v \in V_i \cap V_j$ for some $i \neq j$; similarly an edge $e \in E$ is *shared* if $e \in E_i \cap E_j$. For *node slots*:

$$\sigma_V(v, g_i) = \begin{cases} \deg_G(v) - \deg_{g_i}(v) & \text{if } g_i \text{ is acyclic or a simple cycle,} \\ (a, b) & \text{if } g_i \text{ is a fused cycle,} \end{cases} \tag{5}$$

where $(a, b) \in \mathbb{N}^2$ records required connections to fused-cycle vs. non-fused primitives. For *edge slots*, we set $\sigma_E(e) = 1$ if $e \in E_{\text{sh}}$ (an overlap/fusion boundary) and $\sigma_E(e) = 0$ otherwise.

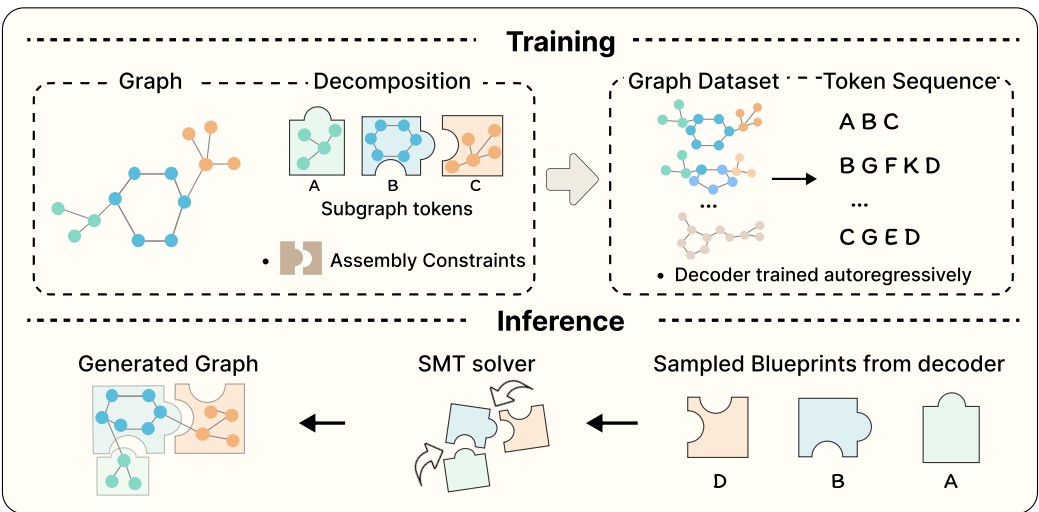

Figure 1: The NSGGM framework: graphs are decomposed into subgraph tokens (top), and a decoder generates token sequences that are assembled into valid graphs (bottom).

**Molecular graph extension.** For molecular graphs, we enrich slots with chemistry-aware attributes: *element type*, *valency*, and *bond order*. Let $\mathcal{E}$ be the set of element types and $\mathcal{B}$ the set of bond orders (e.g., $\mathcal{B} = \{1, 2, 3\}$ for single/double/triple), and define $\mathrm{elem} \colon V \to \mathcal{E}$, $\mathrm{val} \colon V \to \mathbb{N}$, and $\mathrm{bo} \colon E \to \mathcal{B}$. We set

$$\sigma_V(v, g_i) = \big(\mathrm{elem}(v),\, \mathrm{val}(v),\, r_i(v)\big), \qquad r_i(v) \coloneqq \mathrm{val}(v) - \sum_{e \in E_i(v)} \mathrm{bo}(e), \tag{6}$$

where $E_i(v)$ are bonds incident to $v$ within token $g_i$, and we require $r_i(v) \geq 0$ (the *residual valence capacity* to be satisfied by attachments to other tokens). For shared internal bonds, we record their bond order:

$$\sigma_E(e, g_i) \coloneqq \mathrm{bo}(e), \qquad e \in E_i \cap E_{\mathrm{sh}}. \tag{7}$$

We prove that Structural Partitioning yields an overlapping cover of $G$ with cycles and tree components as primitives, and uniquely determined interface slots (Proposition B.1, Appendix B.1).

**Vocabulary and blueprint.** An annotated primitive (token) is

$$t_i = (g_i,\, \sigma_V|_{V_i},\, \sigma_E|_{E_i}). \tag{8}$$

The *blueprint* for $G$ is the multiset $S_G = \{t_i\}_{g_i \in \mathcal{P}(G)}$, and the global *vocabulary* $\mathcal{V}$ is the set of unique token types (up to isomorphism) discovered across $\mathcal{G}$. During assembly, shared nodes are uniquely relabeled within each token so the solver treats them as distinct local copies; whenever two copies are identified, their interface slots must be compatible—i.e., type fields match exactly (same element, valency; same bond order on shared edges) and consumptive fields are satisfied (all residual valence is exactly used with no deficit or surplus).

## 3.2 Assembly constraints for graph synthesis

Given a blueprint $S' = \{t_1, \ldots, t_k\}$ of slotted primitives $t_i = (P_i, \sigma_V|_{V_i}, \sigma_E|_{E_i})$, determine a *merging of nodes* across primitives such that (i) matched node and edge slots are compatible (type-equality and residual-capacity satisfaction), and (ii) the induced identifications yield a consistent graph $G'$ respecting all connections. This is encoded as an SMT constraint-satisfaction problem with variables for node identifications and constraints enforcing slot compatibility and global consistency.

**Decision variables and structural objective.** Let $V_{\mathrm{slots}}$ be the set of all slotted nodes across tokens in $S'$. For each $v \in V_{\mathrm{slots}}$, introduce an integer decision variable $z_v$. Two nodes are merged in $G'$ iff

their variables coincide:

$$\text{merge}(v_i, v_j) \iff z_{v_i} = z_{v_j}. \tag{9}$$

The solver assigns all $\{z_v\}$ to satisfy a set of structural constraints $\phi_{\text{struct}} = \phi_{\text{hard}} \cup \phi_{\text{soft}}$.

**Hard constraints ($\phi_{\text{hard}}$).** These are inviolable and enforce topological integrity and interface consistency. *(i) Subgraph integrity.* Nodes from the same primitive never merge. For any $P_i = (V_i, E_i)$ and distinct $v_a, v_b \in V_i \cap V_{\text{slots}}$,

$$z_{v_a} \neq z_{v_b}. \tag{10}$$

*(ii) Edge–slot matching.* A structural edge slot must be realized by a matching slot in a different primitive with consistent endpoint mergers. If $e = (u, v) \in E_i$ has a structural edge slot (i.e., $\sigma_E^{\text{str}}(e, P_i) = 1$), then there exists $j \neq i$ and $e' = (u', v') \in E_j$ with $\sigma_E^{\text{str}}(e', P_j) = 1$ such that

$$(z_u = z_{u'} \wedge z_v = z_{v'}) \vee (z_u = z_{v'} \wedge z_v = z_{u'}). \tag{11}$$

*(iii) Type equality on matched copies.* Whenever local copies of the same global item are merged, type fields agree:

$$\text{if } z_{v_i} = z_{v_j} \text{ then } \sigma_V^{\text{type}}(v_i, P_i) = \sigma_V^{\text{type}}(v_j, P_j), \tag{12}$$

where $\sigma_V^{\text{type}}$ includes discrete identifiers used by the domain (e.g., cycle/fused flags; for molecules this will include element/nominal valency—see below).

**Soft constraints ($\phi_{\text{soft}}$).** Soft constraints guide the solver toward plausible assemblies via a penalty $L(\mathbf{z}) = \sum_{v \in V_{\text{slots}}} \ell(v, \mathbf{z})$, minimized subject to $\phi_{\text{hard}}$.

*(i) Integer structural slots.* For a node $v$ with structural "stub" count $s(v) \in \mathbb{N}_0$ (from $\sigma_V^{\text{str}}$), let $C(v, \mathbf{z}) = \{v' \in V_{\text{slots}} : z_{v'} = z_v\}$ be its merge class. Write $\text{prim}(v')$ for the (unique) primitive containing $v'$. Penalize unmet/excess stubs by

$$\ell(v, \mathbf{z}) = \left| s(v) - \sum_{v' \in C(v, \mathbf{z}) \setminus \{v\}} \deg_{\text{prim}(v')}(v') \right|. \tag{13}$$

*(ii) Fused-cycle tuple slots.* If $v$ has $\sigma_V^{\text{str}}(v, P_i) = (a, b)$ (required counts of fused-cycle vs. non-fused neighbors), let

$$C_{\text{cyc}}(v, \mathbf{z}) = \{v' \in C(v, \mathbf{z}) : \text{prim}(v') \text{ is fused-cycle}\}, \quad C_{\text{oth}}(v, \mathbf{z}) = C(v, \mathbf{z}) \setminus (C_{\text{cyc}}(v, \mathbf{z}) \cup \{v\}), \tag{14}$$

and define

$$\ell(v, \mathbf{z}) = \big| a - (|C_{\text{cyc}}(v, \mathbf{z})| - 1) \big| + \big| b - |C_{\text{oth}}(v, \mathbf{z})| \big|. \tag{15}$$

**Molecular graph constraints.** For molecular graphs, interface compatibility additionally enforces chemical validity. Let $\text{elem} : V \to \mathcal{E}$, $\text{val} : V \to \mathbb{N}_0$, $\text{bo} : E \to \mathcal{B}$, and for each local copy $v \in V_i$ define its residual from Section 3.1 as

$$r_i(v) = \text{val}(v) - \sum_{e \in E_{P_i}(v)} \text{bo}(e) \geq 0, \qquad E_{P_i}(v) := \{e \in E_i : e \text{ incident to } v \text{ in } P_i\}. \tag{16}$$

We introduce inter-token attachments $A \subseteq V_{\text{slots}} \times V_{\text{slots}}$ with bond-order variables $b(u, v) \in \mathcal{B}$ active only when $z_u = z_v$ indicates a merge across primitives. The following are *hard*:

*(i) Element consistency.* If $z_{v_i} = z_{v_j}$ then $\text{elem}(v_i) = \text{elem}(v_j)$.

*(ii) Valency balance.* For each merged atom represented by any $v$, the total *inter-token* bond order equals the sum of residual capacities contributed by its copies:

$$\sum_{(v,w) \in A} b(v, w) = R(v) := \sum_{i: v \in V_i} r_i(v). \tag{17}$$

*(iii) Shared-edge order consistency.* If an edge $e$ appears in multiple primitives ($e \in E_i \cap E_j \subseteq E_{\text{sh}}$), then $\text{bo}(e)$ is identical across copies: $\text{bo}_i(e) = \text{bo}_j(e)$.

*Correctness guarantees.* For any blueprint $S'$, there exists a feasible assignment that reassembles each training graph from its own tokens (Corollary B.2). Moreover, any assignment satisfying $\phi_{\text{hard}}$ yields a well-defined simple graph; in the molecular case it also enforces element consistency, unique bond orders, and exact valency (Theorem B.3). Together, these imply correctness-by-construction and completeness for training graphs (Corollary B.4). Proofs are provided in Appendix B.1.

**User-controlled constraints.** A key advantage of our approach is extensibility: users can supply domain constraints $\phi_{\text{user}}$ that are injected alongside $\phi_{\text{hard}}$ (and optionally as weighted soft terms in $L$) without changing the solver. These clauses can target structure (e.g., ring-count bounds, forbidden substructures, connectivity/diameter limits) or chemistry (e.g., element/valency guards, required/forbidden functional groups, bond-order budgets), and may be scoped to specific token types or regions. The SMT solver then returns $\{z_v\}$ (and, for molecules, $\{b(u,v)\}$) that satisfy all hard constraints while minimizing $L(\mathbf{z})$, yielding a deterministic assembly of $S'$ into $G'$.

### 3.3 Neural blueprint bodeling and graph generation

Under our decomposition, each graph is represented by a finite sequence of annotated subgraph tokens (a *blueprint*). We model the blueprint distribution with an autoregressive decoder, then invoke symbolic assembly to recover a unique graph irrespective of the token order.

**Blueprints as sequences.** Let a blueprint be an ordered sequence $S = (t_1, \ldots, t_k, \texttt{<eos>})$ with tokens $t_\ell = (P_{i_\ell}, \sigma_V|_{V_{i_\ell}}, \sigma_E|_{E_{i_\ell}}) \in \mathcal{V}$. We parameterize

$$P_\theta(S) = \prod_{\ell=1}^{k+1} P_\theta(t_\ell \mid t_{<\ell}), \qquad t_{k+1} \equiv \texttt{<eos>}. \tag{18}$$

Although $S$ is ordered for modeling convenience, the downstream assembly in Section 3.2 is *order-invariant*: distinct permutations of the same multiset $\{t_\ell\}$ produce the same assembled graph $G'$.

**Training.** Let $f(G)$ map a graph to *a* canonical token sequence (e.g., via a stable traversal of $\mathcal{P}(G)$); then

$$\max_\theta \ \mathbb{E}_{G \sim P_\mathcal{D}}\Big[ \log P_\theta\big(S = f(G)\big) \Big] = \max_\theta \ \mathbb{E}_G \sum_\ell \log P_\theta\big(t_\ell \mid t_{<\ell}\big). \tag{19}$$

To reduce sensitivity to ordering, we optionally train with randomized yet valid permutations $\pi$ of $f(G)$ that preserve token multiset and local interfaces:

$$\max_\theta \ \mathbb{E}_G \ \mathbb{E}_\pi \Big[ \log P_\theta\big(\pi \circ f(G)\big) \Big]. \tag{20}$$

During teacher forcing, we apply a *feasibility mask* $m_\ell \in \{0,1\}^{|\mathcal{V}|}$ that zeros out tokens whose structural slots cannot be satisfied given $t_{<\ell}$ (computed from $\sigma_V^{\text{str}}, \sigma_E^{\text{str}}$):

$$P_\theta(t_\ell \mid t_{<\ell}) \propto \text{softmax}(\mathbf{h}_\ell) \odot m_\ell, \qquad m_\ell(t) = \mathbf{1}\big[\text{locally feasible w.r.t. } t_{<\ell}\big]. \tag{21}$$

Training is summarized in Algorithm 2 (see Appendix B.2).

**Inference and user control.** At test time, we generate $S_{\text{candidate}}$ with constrained decoding:

$$t_\ell \sim P_\theta(\cdot \mid t_{<\ell}) \text{ with mask } m_\ell, \quad \text{(top-}k\text{/nucleus sampling, temperature } \tau, \text{ or beam search)}. \tag{22}$$

Users can steer generation by (i) providing a partial prefix $S_{\text{partial}}$ (motif conditioning), (ii) enforcing *hard* clauses $\phi_{\text{user}}^{\text{hard}}$ (e.g., ring-count bounds, forbidden fragments, distance limits), and/or (iii) weighting *soft* preferences in the decoding score. The resulting candidate blueprint $S_{\text{candidate}}$ and combined constraints $\Phi = \phi_{\text{struct}} \cup \phi_{\text{user}}$ are passed to the SMT stage, which introduces identification variables $\{z_v\}$ for slotted nodes (and bond-order variables $\{b(u,v)\}$ for molecules) and solves for a model that satisfies all hard constraints while minimizing $L(\mathbf{z})$. This yields a concrete graph $G'$ that is *correct by construction* and independent of the generation order (see Algorithm 3 in Appendix B.2).

## 4 Evaluation

We compare NSGGM against state-of-the-art graph generators, including Set2GraphVAE (Vignac & Frossard, 2021), SPECTRE (Martinkus et al., 2022), GraphNVP (Madhawa et al., 2019a), GDSS (Jo et al., 2022a), GraphRNN (You et al., 2018), GRAN (Liao et al., 2019a), JT-VAE (Jin et al., 2018a), NAGVAE (Kwon et al., 2019), GraphINVENT (Mercado et al., 2021b), DiGress and its continuous (Gaussian) variant CONGRESS (Vignac et al., 2023b), and UniGEM (Feng et al., 2025). We report results on the molecular benchmarks QM9 (Wu et al., 2018), MOSES (Polykovskiy et al., 2020), and GuacaMol (Brown et al., 2019); dataset details are in Appendix C. Results on the non-molecular benchmark of Martinkus et al. (2022) are presented in Section F. Further experimental details, including implementation specifics, are provided in Appendix D.

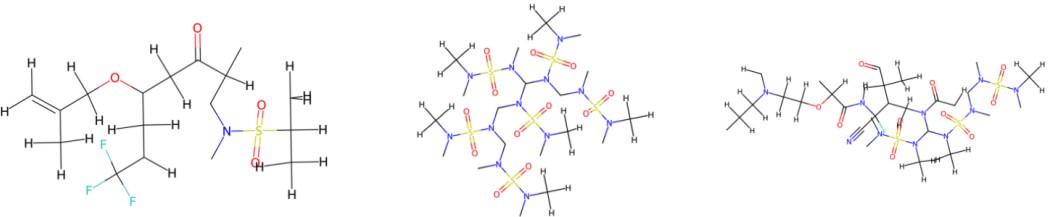

(a) Prompt-based completion from user-supplied partial structures (MOSES). Top row: user prompts (scaffolds). Middle row: the sampler's completion of the remaining sequence. Bottom row: the final molecule assembled by the SMT solver under standard chemistry constraints. Red overlays mark the original user prompt.

(b) Constraint-driven synthesis to satisfy user-specified properties (MOSES). The three examples are generated under a user-specified constraint—no cycles—by enforcing a global acyclicity constraint at solve time.

Figure 2: NSGGM provides fine-grained, user-steerable control that existing methods do not support.

### 4.1 GENERATION WITH FINE-GRAINED, USER-STEERABLE CONTROL

Because the solver is symbolic, the same interface supports strong structural specifications *without retraining*. Beyond feasibility checks, users can declaratively require or forbid patterns (e.g., cycles, ring sizes, attachment points, substructure counts), and these constraints are enforced exactly during assembly rather than approximated via conditioning. Given a user-supplied scaffold, the sampler completes the remaining sequence and the SMT solver assembles a final molecule under standard chemistry constraints; Figure 2a shows two examples that remain deterministically anchored to the prompt while still permitting diverse, valid completions. Global properties can likewise be imposed at solve time; in Figure 2b, a user-specified no-cycles constraint eliminates ring-forming actions, producing strictly acyclic molecules by construction. This separation of concerns, learned proposal for vocabulary-like fragments, and symbolic checking for structure, yields precise control with minimal engineering overhead. In sum, scaffold anchoring and global constraints are achieved at inference time—*without retraining*—whereas purely neural baselines typically require latent-space tricks, auxiliary conditioning heads, or fine-tuning, and still cannot guarantee adherence or validity.

### 4.2 SMALL MOLECULE GENERATION

We evaluate on QM9 (Wu et al., 2018) (80%/10%/10% train/val/test) and report RDKit-based *Validity* and *Unique* over 10k samples with 95% confidence intervals (five runs), as well as training time to

Table 1: Molecule generation on QM9 (implicit hydrogen). "Training time" denotes the time to reach 99% validity. On small graphs, NSGGM matches baseline performance while training faster.

| Method | Valid | Unique | Training time (h) |
|---|---|---|---|
| Dataset | 99.3 | 100 | – |
| Set2GraphVAE | 59.9 | 93.8 | – |
| SPECTRE | 87.3 | 35.7 | – |
| GraphNVP | 83.1 | **99.2** | – |
| GDSS | 95.7 | 98.5 | – |
| ConGress | $98.9 \pm 0.1$ | $96.8 \pm 0.2$ | 7.2 |
| DiGress | $99.0 \pm 0.1$ | $96.2 \pm 0.1$ | 1.0 |
| UniGEM | 95.0 | 93.2 | – |
| NSGGM (ours) | $\mathbf{100.0 \pm 0.0}$ | $96.4 \pm 0.2$ | **0.5** |

Table 2: Molecule generation on QM9 (explicit hydrogens). NSGGM attains perfect *Validity* and stability (both *Atom* and *Mol*), while diffusion-based baselines retain higher *Unique/Novelty*. This reflects a trade-off: our exact assembly enforces strict chemical constraints, yielding correctness by construction but a tighter sample distribution. Arrows indicate higher is better.

| Model | Valid ↑ | Unique ↑ | Atom stability ↑ | Mol stability ↑ |
|---|---|---|---|---|
| Dataset | 97.8 | 100 | 98.5 | 87.0 |
| ConGress | $86.7 \pm 1.8$ | $\mathbf{98.4 \pm 0.1}$ | $97.2 \pm 0.2$ | $69.5 \pm 1.6$ |
| DiGress (uniform) | $89.8 \pm 1.2$ | $97.8 \pm 0.2$ | $97.3 \pm 0.1$ | $70.5 \pm 2.1$ |
| DiGress (marginal) | $92.3 \pm 2.5$ | $97.9 \pm 0.2$ | $97.3 \pm 0.8$ | $66.8 \pm 11.8$ |
| DiGress (marg. + features) | $95.4 \pm 1.1$ | $97.6 \pm 0.4$ | $98.1 \pm 0.3$ | $79.8 \pm 5.6$ |
| NSGGM (ours) | $\mathbf{100.0 \pm 0.0}$ | $95.5 \pm 0.3$ | $\mathbf{100.0 \pm 0.0}$ | $\mathbf{100.0 \pm 0.0}$ |

99% validity (Table 1). We train and evaluate under both hydrogen conventions—implicit hydrogen (Table 1) and explicit hydrogen (Table 2). Graph-level NLL is intractable, so we do not report it. On implicit hydrogen, NSGGM attains $100.0 \pm 0.0$ *Validity* with competitive *Unique* ($96.4 \pm 0.2$) while reaching the 99% validity threshold fastest (0.5 h), outperforming DiGress (1.0 h) and ConGress (7.2 h). On explicit hydrogen, NSGGM achieves perfect *Validity* and perfect *Atom/Mol* stability (both $100.0 \pm 0.0$), whereas diffusion baselines retain slightly higher *Unique*; this reflects a trade-off, as exact assembly enforces strict chemical constraints and thus produces a tighter sample distribution with guaranteed correctness.

Our framework is explicitly designed to incorporate user-specified structural checks; accordingly, in this and all subsequent evaluations, we enable standard chemistry constraints. Precise settings appear in Appendix D. Training is faster because only a lightweight decoder proposes vocabulary-like substructures for the solver. Overall, NSGGM matches or exceeds state-of-the-art validity while providing deterministic constraint satisfaction that purely neural models cannot guarantee.

### 4.3 MOLECULE GENERATION AT SCALE

We scale to GuacaMol (Brown et al., 2019) (Table 3) and MOSES (Polykovskiy et al., 2020) (Table 4), which contain larger and more diverse molecules. To our knowledge, NSGGM is the first interpretable neuro-symbolic generator (autoregressive proposal + SMT enforcement) to operate at this scale. Metric definitions and experimental details appear in Appendix C.

On GuacaMol, NSGGM attains perfect *Validity* and strong dataset-level fit (KL = 81.15, FCD = 46.02) while maintaining high *Novel* (89.47%). Its *Unique* (60.53%) is lower than sequence/one-shot baselines, reflecting tighter structural control and exact constraint satisfaction. On MOSES, NSGGM again achieves 100% *Val* with perfect *Novel*, but trails diffusion and sequence models on distributional similarity metrics (e.g., FCD) and scaffold diversity. This trade-off is expected: symbolic assembly guarantees chemical correctness and user-enforceable constraints by construction, at the cost of a narrower sample distribution under conservative vocabularies and constraints.

Table 3: Molecule generation on GuacaMol. NSGGM attains 100% Valid with strong KL/FCD and high Novel, trading Unique for strict, constraint-respecting assembly and explicit user control.

| Model | Class | Valid ↑ | Unique ↑ | Novel ↑ | KL div ↑ | FCD ↑ |
|---|---|---|---|---|---|---|
| JT-VAE | Fragment | 95.9 | 100 | 91.2 | 99.1 | 91.3 |
| LSTM | SMILES | 95.9 | 100 | 91.2 | 99.1 | 91.3 |
| NAGVAE | One-shot | 92.9 | 95.5 | 100 | 38.4 | 0.9 |
| MCTS | One-shot | 100 | 100 | 95.4 | 82.2 | 1.5 |
| ConGress | One-shot | 0.1 | 100 | 100 | 36.1 | 0.0 |
| DiGress | One-shot | 85.2 | 100 | 99.9 | 92.9 | 68.0 |
| NSGGM (ours) | Neuro-Symbolic | 100 | 60.53 | 89.47 | 81.15 | 46.02 |

Table 4: Molecule generation on MOSES. NSGGM reaches 100% Val and Novel; conservative Unique/FCD reflect guaranteed validity via symbolic assembly.

| Model | Class | Val ↑ | Unique ↑ | Novel ↑ | Filters ↑ | FCD ↓ | SNN ↑ | Scaf ↑ |
|---|---|---|---|---|---|---|---|---|
| VAE | SMILES | 97.7 | 99.8 | 69.5 | 99.7 | 0.57 | 0.58 | 5.9 |
| JT-VAE | Fragment | 100 | 100 | 99.9 | 97.8 | 1.00 | 0.53 | 10 |
| GraphINVENT | Autoreg. | 96.4 | 99.8 | – | 95.0 | 1.22 | 0.54 | 12.7 |
| ConGress | One-shot | 83.4 | 99.9 | 96.4 | 94.8 | 1.48 | 0.50 | 16.4 |
| DiGress | One-shot | 85.7 | 100 | 95.0 | 97.1 | 1.19 | 0.52 | 14.8 |
| NSGGM (ours) | Neuro-Symbolic | 100.0 | 84.8 | 100.0 | 43.5 | 41.26 | 0.25 | 0.0 |

## 5 RELATED WORK

The prevailing family of graph generative methods is diffusion. Earlier "continuous-noise" approaches add Gaussian noise and denoise real-valued tensors: Niu et al. (2020) threshold continuous scores to recover edges; Jo et al. (2022b) extend this to node and edge attributes. Discrete diffusion instead corrupts categorical node/edge types and denoises back to graphs. Vignac et al. (2023a) jointly corrupt and denoise edges and types with a graph transformer, achieving strong coverage and fidelity directly in graph space. Feng et al. (2025) unify molecular generation and property prediction, activating property-aware objectives late in the schedule (after a scaffold emerges) to reduce conflicts and improve sample quality and predictive accuracy. Concurrently, Haefeli et al. (2023) study unattributed graphs and also find discrete diffusion effective. For 3D molecules, Trippe et al. (2022) and Hoogeboom et al. (2022) generate atomic coordinates (point clouds), whereas Xu et al. (2022) and Jiang et al. (2024) perform conformation generation given a fixed graph. Beyond diffusion, non-autoregressive VAEs, GANs, and flows have been explored (Zhu et al., 2022; Madhawa et al., 2019b; Liu et al., 2018b; Luo et al., 2021), but they typically lag strong autoregressive models (Liao et al., 2019b; Mercado et al., 2021c) and motif-based methods (Jin et al., 2020a; Maziarz et al., 2022) that encode domain knowledge. In contrast to diffusion's largely *implicit* conditioning, our neural-symbolic approach offers *explicit*, human-readable rules that clarify *why* substructures assemble and enable fine-grained, verifiable constraints during construction.

## 6 CONCLUSION

We introduce Neuro-Symbolic Graph Generative Modeling (NSGGM), a novel approach that reframes graph generation as sequence modeling followed by constraint-satisfaction solving. NSGGM learns subgraph tokens with typed interfaces, emits a single-pass neural blueprint, and compiles it into a graph via an SMT solver. This yields controllability and verifiability: it eliminates iterative refinement, supports user-steerable constraints and partial prompts, and ensures validity by construction with auditable reasoning. Across molecular and general graph benchmarks, NSGGM attains state-of-the-art quality while enabling fine-grained, user-steerable control. Future work includes scaling token vocabularies to richer, heterogeneous domains; strengthening the constraint system with higher-level templates, domain priors, and more efficient compilation; and leveraging solver feedback to adapt tokenization and decoding. We conclude that NSGGM offers a practical path to trustworthy, controllable, and interpretable graph generation for safety-critical applications.

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

## A    FURTHER DECOMPOSITION OF ACYCLIC COMPONENTS

Each primitive $g_i \in \mathcal{P}(G)$ is a subgraph with node set $V_i$ and edge set $E_i$. For the acyclic part, we refine connected components into recurring *tree motifs* using standard steps:

1. **BC decomposition:** Decompose at articulation points via a block–cut (BC) decomposition (Tarjan, 1972).

2. **Path compression:** Optionally compress degree-2 chains (a standard tree reduction).

3. **Frequent subtree mining:** Canonicalize subtrees and select frequent patterns by minimum support, following frequent (sub)tree mining frameworks (Asai et al., 2003).

4. **Canonical labeling:** Use canonical labeling for isomorphism handling (McKay & Piperno, 2014).

Residual fragments that do not meet the support threshold remain as minimal trees (or single edges).

---

**Algorithm 1:** Refine Acyclic Components into Tree Motifs

---
**Input:** Graph $G = (V, E)$, acyclic edge set $E_A$, minimum support $\tau$
**Output:** Tree-motif tokens $\mathcal{T}_{\text{tree}}$, residual fragments $\mathcal{R}_{\text{tree}}$

1 **Function** RefineAcyclic$(G, E_A, \tau)$
2     $G_A \leftarrow (V, E_A)$                           `// Acyclic remainder subgraph`
3     $\mathcal{C} \leftarrow$ ConnectedComponents$(G_A)$
4     $\mathcal{T}_{\text{tree}} \leftarrow \varnothing$, $\mathcal{R}_{\text{tree}} \leftarrow \varnothing$
5     **foreach** $C \in \mathcal{C}$ **do**
       `// (1) BC decomposition and optional path compression`
6        $(\mathcal{B}, \mathcal{A}) \leftarrow$ BlockCutDecompose$(C)$    `// Blocks` $\mathcal{B}$ `and articulation points` $\mathcal{A}$
7        $\mathcal{B} \leftarrow \{$CompressDegTwoChains$(B) \,|\, B \in \mathcal{B}\}$    `// Contract degree-2 chains within blocks`
       `// (2) Subtree enumeration and canonicalization`
8        $\mathcal{S} \leftarrow \bigcup_{B \in \mathcal{B}}$ EnumerateSubtrees$(B)$       `// Enumerate candidate subtrees (bounded depth/size)`
9        $\mathcal{K} \leftarrow \{$CanonicalizeSubtree$(S) \,|\, S \in \mathcal{S}\}$        `// Canonical forms for isomorphism handling`
       `// (3) Support counting (frequent subtree mining)`
10       supp$(\cdot) \leftarrow$ CountSupport$(\mathcal{K}$, over all $C' \in \mathcal{C})$
11       $\mathcal{F} \leftarrow \{K \in \mathcal{K} \,|\, \text{supp}(K) \geq \tau\}$              `// Frequent patterns`
       `// (4) Tokenization and interface slots`
12       **foreach** $K \in \mathcal{F}$ **do**
13          $g \leftarrow$ RepresentativeSubtree$(K)$
14          $\sigma_V, \sigma_E \leftarrow$ ComputeSlots$(g)$       `// Node/edge slots for interfaces`
15          $\mathcal{T}_{\text{tree}} \leftarrow \mathcal{T}_{\text{tree}} \cup \{$AssembleToken$(g, \sigma_V, \sigma_E)\}$
       `// (5) Residuals: pieces not covered by frequent motifs`
16       $\mathcal{U} \leftarrow$ MaximalUncoveredFragments$(C, \mathcal{F})$            `// after covering by` $\mathcal{F}$
17       $\mathcal{R}_{\text{tree}} \leftarrow \mathcal{R}_{\text{tree}} \cup \mathcal{U}$                `// minimal trees / single edges`
18     **return** $\mathcal{T}_{tree}$, $\mathcal{R}_{tree}$

---

## B    THE NSGGM METHOD

### B.1    PROOFS

**Proposition B.1** (Legitimacy of Structural Partitioning). *Let $G = (V, E)$ be any finite simple graph and let $\mathcal{C}(G) = \{c_1, \ldots, c_p\}$ be a minimum cycle basis with edge sets $E(c_j)$. Define*

$$E_{\mathcal{C}} = \bigcup_{j=1}^{p} E(c_j), \qquad E_A = E \setminus E_{\mathcal{C}}, \qquad G_A = (V, E_A).$$

*Let $\mathcal{P}(G) = \mathcal{C}(G) \cup \text{Components}(G_A)$ and enumerate $\mathcal{P}(G) = \{P_i = (V_i, E_i)\}_{i=1}^{m}$. Then:*

(i) $\{P_i\}$ *is an* overlapping cover *of G:* $\bigcup_i V_i = V$ *and* $\bigcup_i E_i = E$.

(ii) *Every $P_i$ is either a simple cycle $c_j$ or an induced connected acyclic subgraph (a tree component of $G_A$).*

(iii) *If $v \in V_i \cap V_j$ or $e \in E_i \cap E_j$ for $i \neq j$, then the overlap corresponds to a genuine shared item of $G$ (i.e., no spurious duplication).*

(iv) *The blueprint $S_G = \{t_i\}_{i=1}^m$ with tokens $t_i = (P_i, \sigma_V|_{V_i}, \sigma_E|_{E_i})$ is well-defined; in particular, the slot maps $\sigma_V, \sigma_E$ can be computed deterministically from $G$ and $\mathcal{P}(G)$.*

*Proof.* **(i)** By construction, $E = E_{\mathcal{C}} \cup E_A$ and the union is disjoint. The family $\{E(c_j)\}_j$ covers $E_{\mathcal{C}}$, while $\mathrm{Components}(G_A)$ covers $E_A$. Hence $\bigcup_i E_i = E$. Since every edge's endpoints are included, $\bigcup_i V_i = V$ follows.

**(ii)** Each $c_j \in \mathcal{C}(G)$ is a simple cycle by definition of a cycle basis. Deleting $E_{\mathcal{C}}$ breaks all remaining cycles, so $G_A = (V, E_A)$ is acyclic; its connected components are trees.

**(iii)** If an item of $G$ appears in two primitives, it is because that item is simultaneously part of a cycle and of another cycle (fused cycles) or it lies on the interface between a cycle and a tree component (or between fused cycles). In all cases, the overlap is an actual vertex/edge of $G$ present in both subgraphs, not an artifact of the construction.

**(iv)** The slot assignments $\sigma_V, \sigma_E$ are functions of local degrees in $G$ and in each primitive, plus the fused/non-fused tag for cycle primitives (Section 3.1). Since these are determined from $(G, \mathcal{P}(G))$ with no ambiguity, $t_i$ is uniquely defined for each $P_i$. $\square$

**Corollary B.2** (Canonical Witness for Reassembly). *Let $S_G$ be the blueprint of $G$ from Proposition B.1. There exists an assignment $\{z_v\}_{v \in V_{\mathrm{slots}}}$ (take $z_v$ equal to the original global vertex ID of $v$) such that all hard constraints $\phi_{hard}$ in Section 3.2 are satisfied. Thus, every training graph admits a feasible assembly consistent with its own blueprint.*

*Proof.* Assign each local copy $v \in V_i$ the label $z_v := \mathrm{id}_G(v)$ given by its vertex in $G$. Subgraph integrity holds because distinct local copies within the same primitive correspond to distinct vertices. Edge–slot matching holds since each shared edge of $G$ appears with the same endpoints across primitives. Type-equality is immediate because copies refer to the same underlying item of $G$. Hence $\phi_{\mathrm{hard}}$ is satisfied. $\square$

**Theorem B.3** (Soundness of Assembly Under $\phi_{\mathrm{hard}}$). *Fix a blueprint $S' = \{t_1, \ldots, t_k\}$ and any assignment $\{z_v\}_{v \in V_{\mathrm{slots}}}$ that satisfies all hard constraints $\phi_{hard}$ of Section 3.2. Define an equivalence relation on slot-copies by $v \sim v' \iff z_v = z_{v'}$ and let $[v]$ denote its classes. Construct a graph $G' = (V', E')$ with $V' = \{[v] : v \in V_{\mathrm{slots}}\}$ and with edges induced from primitive edges subject to edge–slot matching. Then:*

(i) *$G'$ is a well-defined simple graph (no self-loops or parallel edges are introduced by merging).*

(ii) *The embedding of each primitive $P_i$ into $G'$ is injective on its vertices and edges (subgraph integrity).*

(iii) *If two local copies are merged, their discrete type fields match exactly (type consistency).*

*If, in addition, $S'$ is molecular and the molecular hard clauses hold (element consistency, shared-edge order consistency, and exact valency balance), then:*

(iv) *Every merged vertex in $G'$ has element type well-defined and unique.*

(v) *For each atom $u \in V'$, the total bond order of edges incident to $u$ equals its prescribed valency; in particular, no deficit or surplus valence occurs.*

(vi) *Any edge $e$ occurring in multiple primitives has a single bond order in $G'$.*

*Consequently, $G'$ is a topologically valid graph; in the molecular case it is chemically valid by construction.*

*Proof.* **(i)** Define $V'$ as the quotient by $\sim$. By *subgraph integrity*, two distinct vertices of the same primitive never merge, so a primitive cannot collapse onto itself. By *edge–slot matching*, whenever a structural edge is realized by merging endpoints from two primitives, its endpoints map to distinct equivalence classes (no self-loop). Parallel edges cannot arise because matched edge-slots are required to pairwise realize the same underlying adjacency; any duplicate realization would violate the matching condition.

**(ii)** Subgraph integrity is precisely the clause $z_{v_a} \neq z_{v_b}$ for distinct $v_a, v_b$ of the same primitive; thus the primitive's vertex set injects into $V'$ and its edges map injectively into $E'$.

**(iii)** This is exactly the type-equality constraint: $z_{v_i} = z_{v_j} \Rightarrow \sigma_V^{\text{type}}(v_i) = \sigma_V^{\text{type}}(v_j)$.

**(iv)–(vi) Molecular case.** Element consistency gives a unique element type per class $[v]$. Shared-edge order consistency ensures any edge $e$ appearing in overlaps has a unique bond order. Finally, valency balance enforces

$$\sum_{w:\ ([v],[w]) \in E'} \text{bo}([v], [w]) \;=\; R(v)$$

for each class $[v]$, where $R(v)$ is the total residual contributed by all local copies of that atom. Since residuals are defined as $\text{val}(v) -$ (intramolecular bond order already inside primitives), this equality guarantees that inter-token bonds exactly saturate the valence: neither deficits nor over-saturation can occur. $\qquad\square$

**Corollary B.4** (Correctness by Construction). *If $S'$ is obtained from a valid graph $G$ via the decomposition of Section 3.1, then by Corollary B.2 there exists a satisfying assignment of $\phi_{hard}$ that reconstructs a graph $G^\star$ isomorphic to $G$. Conversely, by Theorem B.3, any satisfying assignment yields a valid $G'$; in the molecular setting, $G'$ also satisfies element consistency and valency exactly. Hence the* hard *constraints are sound (no invalid outputs) and complete for reassembling training graphs (every training graph has a satisfying witness).*

## B.2 The NSGGM algorithms

This appendix gives concise, implementation-ready pseudocode for training and inference. Algorithm 2 builds the token vocabulary $\mathcal{V}$ and structural constraints $\phi_{\text{struct}}$ from the decomposition (Section 3.1), then trains the autoregressive decoder $M_\theta$ to model blueprint sequences (Equations equation 18–equation 19). We also include an optional permutation augmentation (Eq. equation 20) to reduce order sensitivity and a feasibility mask (Eq. equation 21) that rules out locally impossible next tokens based on slot interfaces ($\sigma_V^{\text{str}}, \sigma_E^{\text{str}}$).

At inference (Algorithm 3), we decode a candidate blueprint $S_{\text{candidate}}$ with the same feasibility mask and user-selected decoding policy (top-$k$/nucleus/beam, temperature $\tau$). We then form an SMT instance from $S_{\text{candidate}}$ and the combined constraints $\Phi = \phi_{\text{struct}} \cup \phi_{\text{user}}$, introducing identification variables $\{z_v\}$ for slotted nodes (and bond-order variables $\{b(u,v)\}$ for molecules). The solver produces a model satisfying all hard clauses and minimizing the soft penalty $L(\mathbf{z})$, yielding a graph $G'$ that is correct by construction and independent of token order.

**Practical notes.** (i) *Masking*: BuildFeasibilityMasks precomputes per-position masks using only local interface checks (degree stubs, fused-cycle tuples), which is linear in the number of candidate tokens at each step. (ii) *User control*: $\phi_{\text{user}}$ may add hard clauses (e.g., ring-count bounds, forbidden motifs) or weighted soft terms that are optimized in the final SMT objective without retraining $M_\theta$. (iii) *Caching*: Since $\phi_{\text{struct}}$ depends only on $\mathcal{V}$, its encoding can be cached across runs.

## C Dataset details

### C.1 Molecular graph datasets

QM9 Wu et al. (2018), MOSES and GuacaMol are molecular datasets, where nodes represent atoms and edges correspond to bonds. Planar dataset Martinkus et al. (2022) is a non-molecular graph.

**Algorithm 2:** NSGGM Training Phase

**Input:** Graph dataset $\mathcal{G} = \{G_1, \ldots, G_m\}$
**Output:** Trained decoder $M_\theta$, token vocabulary $\mathcal{V}$, structural constraints $\phi_{\text{struct}}$

1 **Function** NSGGMTrain($\mathcal{G}$)
2     $\mathcal{V} \leftarrow$ GraphDecomposition($\mathcal{G}$)     `/* Build vocabulary from primitives` $P_i$ `and`
    `interfaces` $(\sigma_V, \sigma_E)$ `*/`
3     $\phi_{\text{struct}} \leftarrow$ ExtractConstraints($\mathcal{V}$)     `/* Slot-based structural constraints */`
4     $\mathcal{S} \leftarrow \{ f(G) : G \in \mathcal{G} \}$     `/* Canonical blueprint sequences` $S = f(G)$ `*/`
5     **if** *perm.-aug.* **then**
6         $\mathcal{S} \leftarrow \mathcal{S} \cup \{\pi \circ f(G) : \pi \text{ valid permutation}\}$     `/* Eq. equation 20 */`
7     $\{m_\ell(S)\} \leftarrow$ BuildFeasibilityMasks($\mathcal{S}, \phi_{\text{struct}}$)     `/* Mask infeasible next tokens;`
    `Eq. equation 21 */`
8     $M_\theta \leftarrow$ TrainDecoderMasked($\mathcal{S}, \{m_\ell\}$)     `/* Maximize Eq. equation 19 with masks`
    `*/`
9     **return** $M_\theta$, $\mathcal{V}$, $\phi_{\text{struct}}$

---

**Algorithm 3:** NSGGM Inference Phase

**Input:** Decoder $M_\theta$, vocabulary $\mathcal{V}$, structural constraints $\phi_{\text{struct}}$, optional user constraints $\phi_{\text{user}}$, optional
    prefix $S_{\text{partial}}$
**Output:** Generated graph $G'$

1 **Function** NSGGMGenerate($M_\theta, \mathcal{V}, \phi_{\text{struct}}, \phi_{\text{user}}, S_{\text{partial}}$)
2     $S \leftarrow S_{\text{partial}}$ **or** $\langle\text{start}\rangle$     `/* Motif-conditioned or unconditional */`
3     **while** *not* `<eos>` **do**
4         $m \leftarrow$ BuildMask($S, \phi_{\text{struct}}$)     `/* Local feasibility mask */`
5         $t \sim P_\theta(\cdot \mid S)$ with mask $m$     `/* Top-`$k$`/nucleus/beam, temperature` $\tau$ `*/`
6         $S \leftarrow S \cup \{t\}$
7     $S_{\text{candidate}} \leftarrow S$
8     $\Phi \leftarrow \phi_{\text{struct}} \cup \phi_{\text{user}}$     `/* Combine hard/soft clauses if provided */`
9     $\varphi \leftarrow$ FormulateSMT($S_{\text{candidate}}, \Phi$)     `/* Introduce` $\{z_v\}$ `(and` $\{b(u,v)\}$ `for`
    `molecules) */`
10     $G' \leftarrow$ SMT_Solve($\varphi$)     `/* Satisfy all hard constraints, minimize` $L(\mathbf{z})$ `*/`
11     **return** $G'$

---

**QM9.** QM9 is a supervised, property-rich benchmark of small organic molecules (up to nine heavy atoms among {C,N,O,F}), each paired with a single low-energy 3D conformation and a suite of quantum-chemical targets (e.g., dipole moment, polarizability, energies). Its compact chemical space and dense labels make it ideal for studying message passing, geometry-aware encoders, and multi-task regression. QM9's strength is label depth (node/edge/3D information and many targets), not breadth of chemotypes; it is less suitable for evaluating large-scale distribution learning.

**MOSES.** MOSES is a large, cleaned corpus for generative modeling drawn from drug-like ZINC subsets with standardized filters (e.g., MW, logP, allowed atom types). It ships with train/test splits and a unified metric suite (validity, uniqueness, FCD, KL, internal diversity), encouraging apples-to-apples comparison across unconditional generators. Unlike QM9, MOSES prioritizes scale and distributional realism over supervised labels; molecules are provided primarily as SMILES without per-atom/bond targets.

**GuacaMol.** GuacaMol is a broad, medicinal-chemistry benchmark that evaluates both distribution learning and goal-directed generation (e.g., multi-objective property optimization, rediscovery). It emphasizes realistic scaffold diversity and challenge tasks rather than supervised labels, providing standardized splits, metric implementations, and leaderboards. Practically, the dataset includes many chemically complex structures (e.g., formal charges, fused/bridged rings), which can stress graph↔string toolchains (see footnote in Table 5).

†**Processing note (GuacaMol):** The corpus contains many chemically complex molecules (e.g., formal charges, fused/bridged rings). In our preprocessing, round-tripping *SMILES* → graph → *SMILES* fails for ∼20% of

| Dataset | Scale | Representation | Supervision | Primary evaluation |
|---------|-------|---------------|-------------|-------------------|
| QM9 | ∼134k (small) | Graph + 3D coords | Many phys. targets | Supervised regression (QC) |
| MOSES | ≫1M (very large) | SMILES (2D) | None (unlabeled) | Gen. metrics (validity/FCD/KL) |
| GuacaMol[†] | ∼1.3–1.6M (large) | SMILES (2D) | None (unlabeled) | Gen. & goal-directed (MPO) |

Table 5: Core, high-signal differences across datasets. We omit overlapping details (e.g., tokenization choices) to emphasize what most affects model/evaluation design.

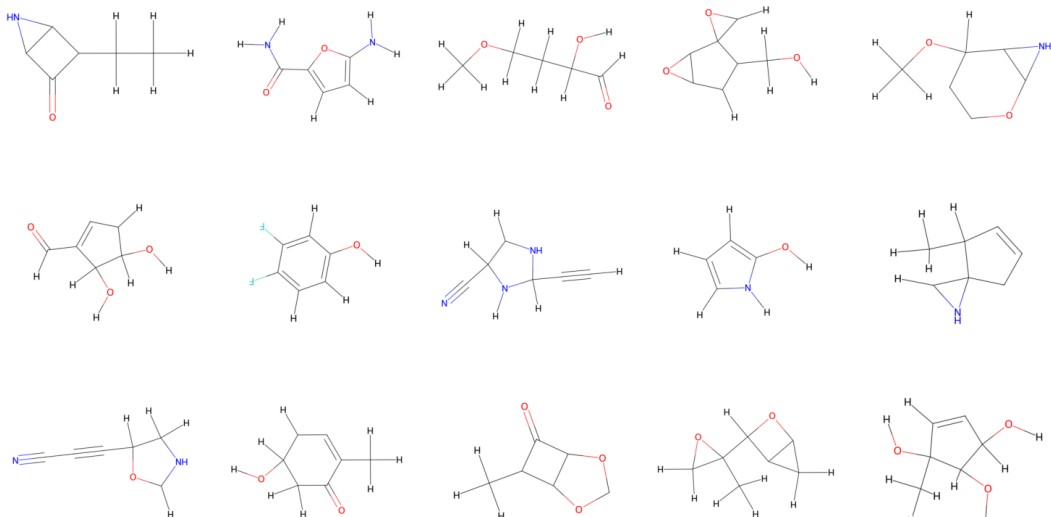

Figure 3: Generated samples for qm9 (Explicit)

training molecules; consequently, some generated graphs cannot be mapped back to *SMILES* for scoring, motivating more robust graph↔string handling.

## D EXPERIMENTAL SETUP

**Hardware.** All experiments are conducted on Ubuntu 22.04 LTS with an AMD EPYC™ 7532 (32 cores), 128 GB RAM, and a single NVIDIA A100 (40 GB). Each dataset is split into training, validation, and testing sets using an 80%/10%/10% ratio unless an official split is provided. All experiments are repeated using three different random seeds to ensure robustness.

**Experiment chemistry constraints.** Across all experiments, we apply a valency-only feasibility check. Bonds contribute their nominal order to the atom valence. Aromatic bonds are counted as 1.5. Default atomic valence limits follow common organic chemistry conventions (i.e, H = 1, C = 4, N = 3 / 5 with charge, ...). Formal charges adjust the allowed valence accordingly.

**Transformer decoder sampler.** For each dataset, we trained the sampler for 20 epochs using the Adam optimizer (learning rate $5 \times 10^{-3}$, batch size 128). The model is a lightweight GPT-style decoder with four layers, four attention heads, and 128 hidden dimensions. At inference time, we employed diverse beam search with nucleus sampling (top-p = 0.99) and top-k sampling (k = 1000). A global frequency of 1.0 is applied to discourage over-sampling frequent fragments.

## E LLM USAGE

Large Language Models (LLMs) were used as a general-purpose assistive tool in the preparation of this work. Specifically, LLMs supported tasks such as refining the clarity of writing, suggesting alternative phrasings, and checking the consistency of technical terminology. They were **not** used for generating research ideas, conducting experiments, or producing original scientific contributions.

972
973
974
975
976
977
978
979
980
981
982
983
984
985
986
987
988
989
990
991
992
993
994
995
996
997
998
999
1000
1001
1002
1003
1004
1005
1006
1007
1008
1009
1010
1011
1012
1013
1014
1015
1016
1017
1018
1019
1020
1021
1022
1023
1024
1025

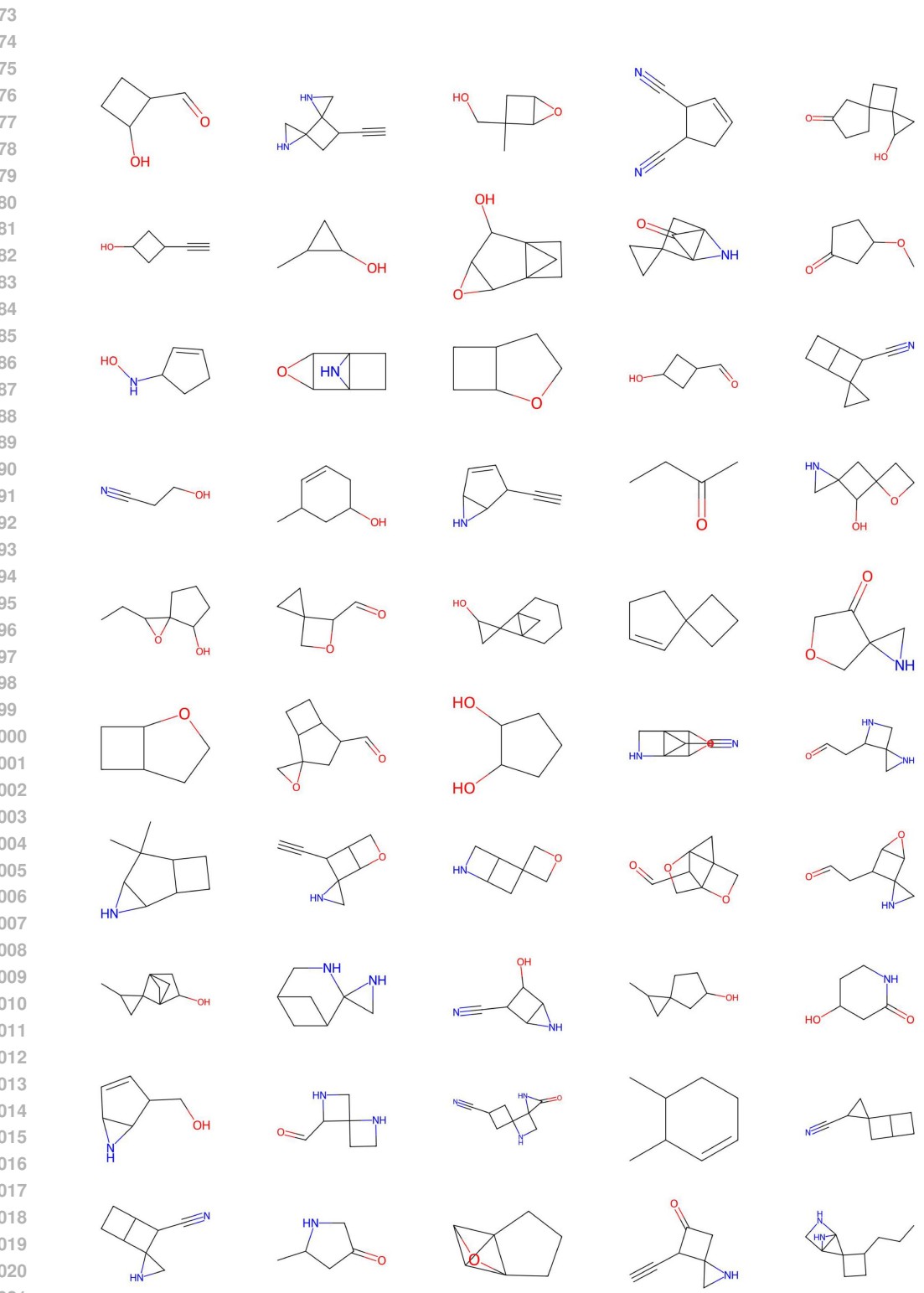

Figure 4: Generated samples for qm9 (implicit)

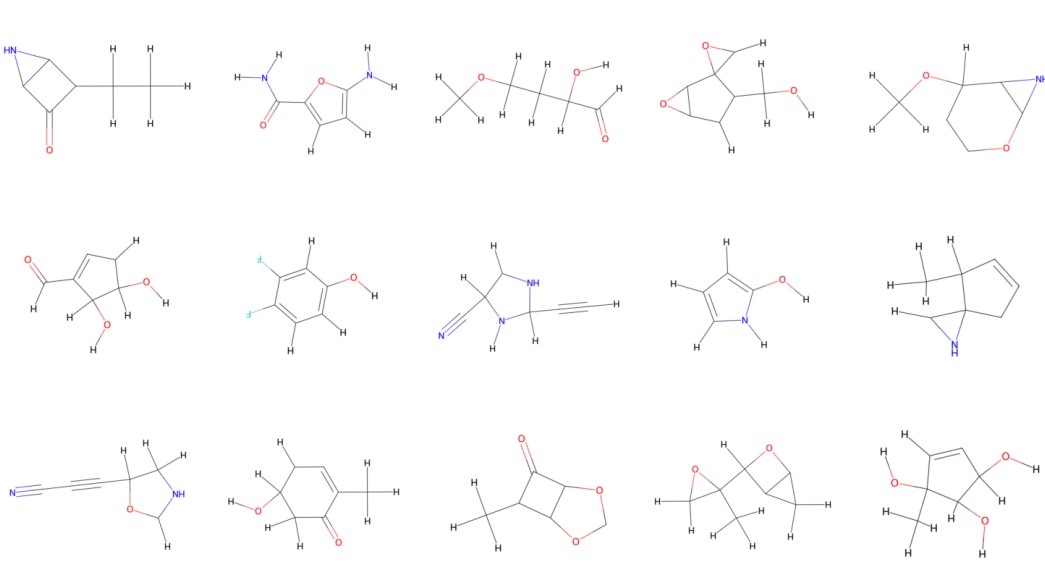

Figure 5: Generated samples for qm9 (Explicit)

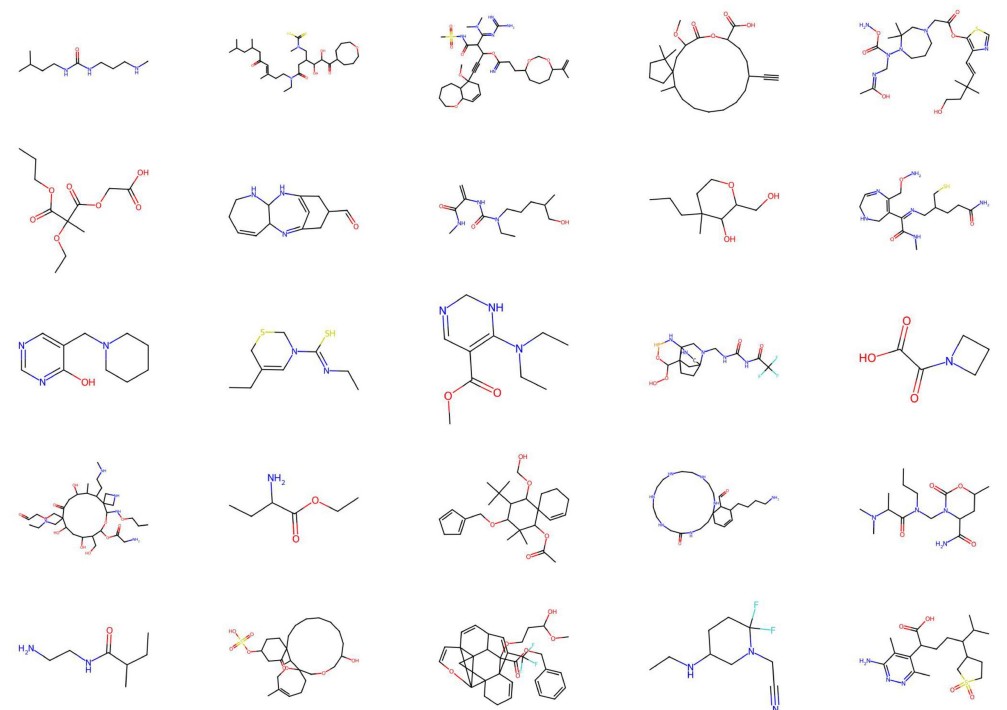

Figure 6: Generated samples for GuacaMol

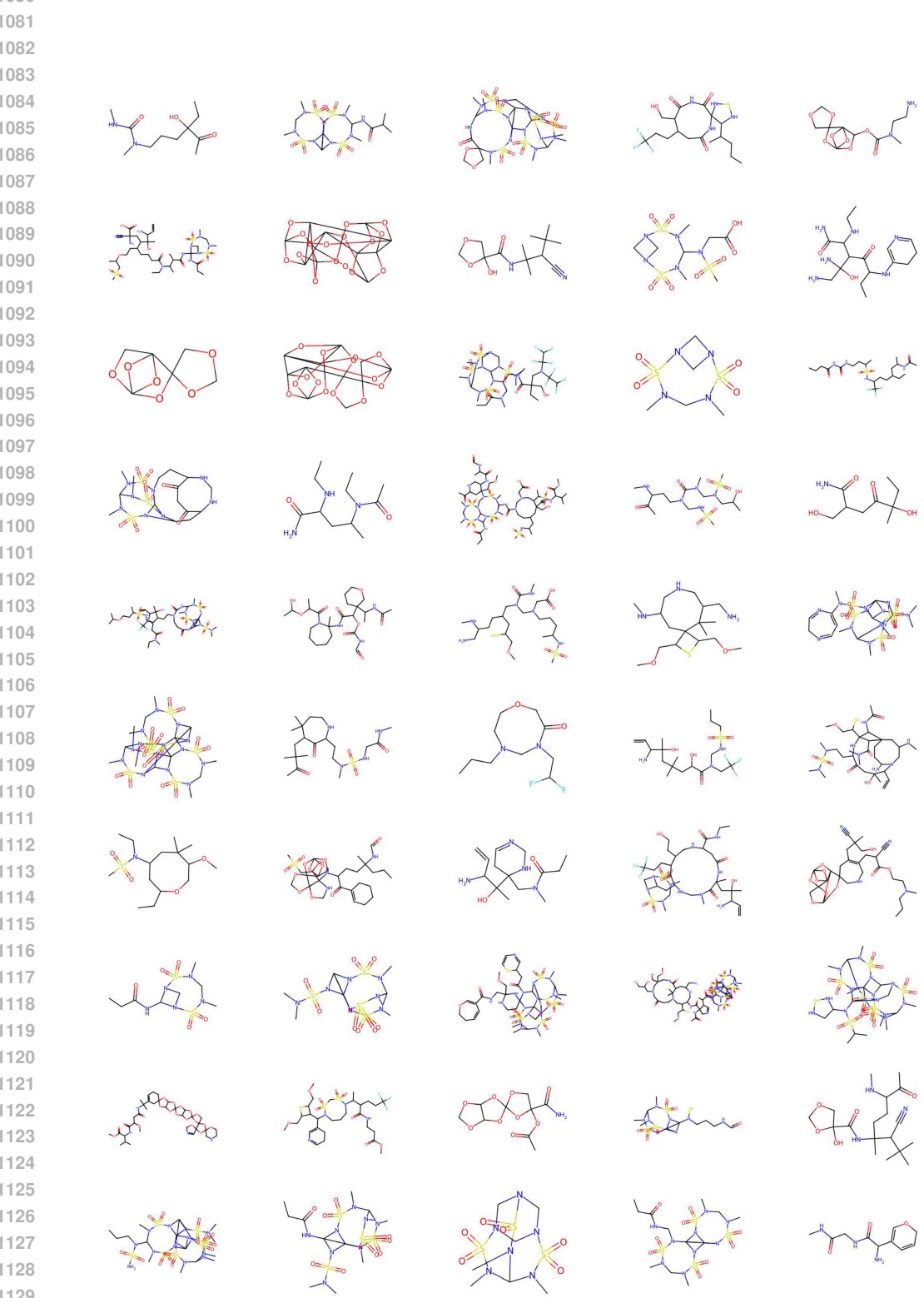

Figure 7: Generated samples for MOSES

All substantive research decisions, analysis, and results presented in this paper are the responsibility of the authors. The authors have carefully reviewed and verified all LLM-assisted text to ensure accuracy and originality.

## F  NON-MOLECULAR GRAPH DATASET

The non-molecular benchmark of Martinkus et al. (2022) consists of two datasets of 200 graphs: (a) stochastic block models (SBM; up to 200 nodes) and (b) planar graphs (64 nodes). Following the protocol, we assess how well generated graphs match degree distributions (Deg), clustering coefficients (Clus), orbit counts (Orb), and the proportion of valid, unique, and novel graphs (V.U.N.).

NSGGM is used in an unconditional setting. The sampler proposes discrete construction sequences, which the symbolic solver assembles under task-specific constraints (e.g., simple connectivity for SBM and planarity for planar graphs). On *SBM*, NSGGM attains the lowest Deg error (1.3) and ties the best Clus (1.5), while remaining competitive on Orb (1.9), yielding the highest V.U.N. (77%). On *Planar*, NSGGM achieves the best Deg (1.3) and Clus (1.1) and competitive Orb (2.0), with V.U.N. 72%—close to DiGress (75%)—indicating strong fidelity to planar structure with minimal loss in novelty. These outcomes illustrate that explicit, verifiable constraints during assembly can improve structural metrics without sacrificing diversity.

Table 6: Unconditional generation on SBM and planar graphs. V.U.N.: valid, unique & novel graphs.

| Model | Deg ↓ | Clus ↓ | Orb ↓ | V.U.N. ↑ |
|---|---|---|---|---|
| *Stochastic block model* | | | | |
| GraphRNN | 6.9 | 1.7 | 3.1 | 5% |
| GRAN | 14.1 | 1.7 | 2.1 | 25% |
| GG-GAN | 4.4 | 2.1 | 2.3 | 25% |
| SPECTRE | 1.9 | 1.6 | **1.6** | 53% |
| ConGress | 34.1 | 3.1 | 4.5 | 0% |
| DiGress | 1.6 | **1.5** | 1.7 | 74% |
| NSGGM (ours) | **1.3** | **1.5** | 1.9 | **77%** |
| *Planar graphs* | | | | |
| GraphRNN | 24.5 | 9.0 | 2508 | 0% |
| GRAN | 3.5 | 1.4 | 1.8 | 0% |
| SPECTRE | 2.5 | 2.5 | 2.4 | 25% |
| ConGress | 23.8 | 8.8 | 2590 | 0% |
| DiGress | 1.4 | 1.2 | **1.7** | **75%** |
| NSGGM (ours) | **1.3** | **1.1** | 2.0 | 72% |

