# OpenReview forum: "Neuro Symbolic Graph Generative Modeling"
_ICLR.cc/2026/Conference — Submitted to ICLR 2026_

### Official Review · Reviewer_xDbv · 2025-10-27

**Soundness:** 2
**Presentation:** 2
**Contribution:** 2
**Rating:** 2
**Confidence:** 3

**Summary:**

The paper is motivated by 3 key challenges faced by diffusion method:
high computational cost, the need for post-hoc filtering to ensure valid outputs, and limited interpretability.

The proposed model, NeurSymb, introduces a neuro-symbolic approach to address these issues.
It decomposes a graph into substructures, treating each as an atomic “word” in a symbolic vocabulary.
These substructures can then be assembled to generate larger graphs, much like constructing sentences from words.

Unlike previous sub-block–based approaches, NeurSymb introduces two innovations:

1 Structure-guided generation through the use of structural prompts provided by users.

2 Validity by construction, ensuring that all generated outputs are guaranteed to be valid without the need for post-processing.


The model employs an autoregressive decoder, which generates intermediate outputs that are subsequently passed to a symbolic solver.
This solver deterministically enforces logical and structural constraints, ensuring correctness.

In the preliminary section, the authors discuss the theoretical foundation based on Satisfiability Modulo Theories (SMT), which underpins the symbolic reasoning process.

NeurSymb’s approach to splitting graphs into subgraphs resembles the JT-VAE framework,
but extends it by allowing multi-tree motifs, offering greater flexibility and structural expressiveness.

The experimental results show that NeurSymb achieves perfect validity (100%),
demonstrating the strength of its symbolic constraints.
However, the model performs poorly on most other metrics,
indicating a trade-off between validity and other aspects of generative quality such as diversity or fidelity.

**Strengths:**

the idea of using neurosymbolic method is well motivated and interesting.

**Weaknesses:**

the idea is not novel, there is much overlap with jt-vae.

experimental results are quite weak, especially on metrics other than validity.

We compare NSGGM against state-of-the-art graph generator"
baselines from 2019/ 2020 are not SOTA, the field moves quickly.

**Questions:**

what are the limitations / tradeoffs of your method?

---

> ### Author Response · Authors · 2025-11-22
>
> We thank the reviewer for the detailed feedback.
>
> **W1: Novelty vs. JT-VAE**
>
> We agree that our subgraph-based approach is related to JT-VAE. However, our method differs in many key ways:
>
> - **Tokenization:** We use a minimum-cycle and multi-tree decomposition with frequent subtree mining, creating a learned vocabulary of multi-tree substructures rather than a fixed set of hand-designed junction trees.
> - **Generation mechanism:** JT-VAE decodes junction trees and molecular graphs via neural methods only. NSGGM separates (i) a neural blueprint proposal decoder (ii) symbolic assembly via SMT with correctness guarantees.
> - **Post-training controllability:** NSGGM allows users to inject new structural constraints (i.e., required/forbidden substructures, ring-count bounds, connectivity constraints) *after training*, enforced exactly by the solver. JT-VAE does not provide such an explicit, editable constraint method.
>
> We will make this distinction to JT-VAE much clearer in the revised version.
>
> **W2: Experimental strength and baselines**
>
> We agree that, in the current configuration, NSGGM trades distributional fit (e.g., FCD, scaffold diversity) for strict validity and symbolic controllability. We will:
>
> - Add ablations on motif granularity and vocabulary size to show how this trade-off can be shifted.
> - Extend the comparison to more recent graph generators where code/evaluation pipelines are available, and explicitly position older baselines as historical reference points rather than SOTA.
>
> **Q1: On limitations / trade-offs**
>
> The main limitations/trade-offs are:
>
> - **Distributional fit vs. constraints:** With the current, relatively coarse vocabulary, NSGGM underperforms strong neural baselines on FCD, KL, and scaffold diversity. This is the price we currently pay for guaranteed validity and strict symbolic constraints; we are exploring finer blueprint substructures and richer priors to mitigate this.
> - **Solver cost:** The SMT step adds overhead per sample and scales with blueprint size, making NSGGM less suitable when only unconstrained, high-throughput sampling is required.
> - **Expressivity tied to vocabulary:** If a structure cannot be expressed by the learned primitives, NSGGM cannot generate it; this is why vocabulary design and coverage analysis are crucial, and we are adding explicit statistics/ablations in our revision.
>
> We will emphasize these trade-offs more explicitly and refine the empirical section to better delineate when one should prefer a neuro-symbolic approach like NSGGM over purely neural methods.

---

### Official Review · Reviewer_DRHy · 2025-10-31

**Soundness:** 2
**Presentation:** 2
**Contribution:** 2
**Rating:** 2
**Confidence:** 3

**Summary:**

The authors propose a new paradigm for generating graphs through Neuro-Symbolic Graph Generative Modelling (NSGGM), which has three main steps: learning a vocabulary of subgraph tokens, autoregressively sampling from that vocabulary, and using an SMT solver to assemble the sampled tokens into a graph. The key claimed advantages are guaranteed validity and fine-grained user-controllable constraints on output graphs.

Unfortunately, the experimental validation has significant weaknesses. First, the evaluation is incomplete and inconsistent: metrics differ across tables without justification, confidence intervals are missing for larger datasets, and critical information such as inference time and vocabulary coverage is not reported. Second, NSGGM shows poor performance on distributional metrics across all larger molecular datasets—for instance, achieving 0.0 scaffold diversity and FCD of 41.26 on MOSES compared to ~1.0 for diffusion methods. Third, while the paper claims controllability as a key advantage over existing methods, this is demonstrated only through qualitative examples without any quantitative comparison to conditional generation baselines.

While guaranteed validity through constraint satisfaction is conceptually appealing, the paper does not make a convincing case for when this justifies the substantial degradation in distributional fit. Overall, NSGGM presents a novel framework but fails to demonstrate practical scenarios where it outperforms existing methods.

**Strengths:**

It is laudable that the paper introduces a new framework (NSGGM) and proposes an alternative to currently prevailing diffusion-based models. The NSGGM framework is well explained and Section 3 is well structured, clearly presenting the decomposition, tokenization, and SMT-based assembly process. The paper provides formal guarantees (Propositions B.1-B.4) for correctness-by-construction, which is a notable theoretical contribution. The demonstrations in Figure 2 showcase a higher level of user control through scaffold completion and constraint-driven synthesis, representing capabilities that are difficult to achieve with purely neural methods.

NSGGM achieves perfect validity (100.0%) across all benchmarks and very high novelty on MOSES (100.0%), though the latter should be interpreted in context with the poor distributional fit metrics (FCD 41.26, 0.0 scaffold diversity).

**Weaknesses:**

My principal concern with the paper is threefold, all aspects related to the evaluation of the framework.

__1. Rigour__
The model is evaluated both on QM9 with and without implicit hydrogen. While this is a good approach, the metrics reported are minimal and not the same, for no clear reason.
- Training time is only reported for implicit hydrogens for example.
-  In table 2, the legend claims diffusion-based baselines "retain higher Unique/Novelty" but novelty is not reported in the table.
- In table 3 and 4, no confidence intervals are specified.
- In table 3, a higher FCD score is indicated to be preferable, in table 4 it is the opposite. This should be corrected or explained.
- Differing metrics are evaluated in table 3 and 4, despite the task, namely larger molecule generation, being the same.

__2. Performance__
The model scores by definition 100.0 on validity and has decent performance on uniqueness/novelty on the larger datasets, albeit poorer than the models it compares to. However, performance on distributional metrics is poor across the board. The Kullback-Leibler divergence is weak and the FCD performance (which from context I assume to be the Fréchet ChemNet Distance) is not competitive. Both SNN and Scaf metrics are poor, and also not explained. On MOSES, the scaffold diversity is 0.0, indicating the model generates from a significantly narrower distribution than the training data. While the authors acknowledge this as a trade-off, they do not provide analysis of vocabulary coverage or demonstrate that the learned tokens can adequately represent the chemical space of the training data.

__3. Evaluation of constraint satisfaction__
The paper's primary claimed advantage over existing methods is fine-grained, user-steerable control through constraint satisfaction. However, this is demonstrated only qualitatively through examples in Figure 2. There is no quantitative evaluation comparing NSGGM's constraint satisfaction capabilities against relevant baselines. Without such evaluation, it is difficult to assess whether the claimed controllability advantage justifies the significant performance degradation on standard metrics.

In addition, ablations on the different novel aspects of the paper would be helpful.

Minor points:

* Metrics like FCD, SNN, Scaf should if not described, at least appear in their spelled out form at some point.
* Atomic and molecular stability cannot be assumed to be known in the graph community and should be defined.
* Figure 2a.: I think it should say "row", not "column" in the legend.

**Questions:**

* In practice, high but not perfect validity is not a major problem in applications of graph generation, since invalid graphs can simply be removed and more generated if novelty is high enough. Can the authors clarify the usefulness of guaranteeing validity?
* Are there ways that the performance of the model can be improved on the chosen benchmarks?
* The authors should discuss how their approach relates to fragment based molecular design and older genetic algorithm approaches such as: https://pubs.rsc.org/en/content/articlelanding/2019/sc/c8sc05372c.
* In figure 6 in the appendix, large ring structures are generated that look very different from what real molecules are expected to look like. Can the author comment on that?
* How does vocabulary size scale with dataset size? Can the vocabulary size be reported for the datasets in the paper?
* How sensitive are results to the minimum support threshold in frequent subtree mining?
* Can you provide quantitative evaluation comparing constraint satisfaction rates and output quality against conditional generation baselines? What complex constraints has NSGGM successfully satisfied that existing methods cannot handle?

Although answering these questions would improve the paper, the main shortcoming lies in the evaluation performance of NSGGM.

---

> ### Author Response · Authors · 2025-11-22
>
> We thank the reviewer for the detailed and constructive feedback. We will respond to the main concerns below.
>
> ---
>
> ### W1: Evaluation rigor / metric reporting
>
> We agree that the evaluation section can be made more consistent and transparent, and we will revise accordingly:
> We will combine the set of metrics reported on QM9 (with/without hydrogens) and on MOSES/GuacaMol, and state clearly when/why a metric is omitted (e.g., not applicable or redundant given others).
> We will report training settings where comparisons are meaningful, including an explicit breakdown of decoder vs. solver time.
> We will add standard deviations for all datasets, not just for QM9.
> We will standardize the convention (lower FCD is better) and explicitly spell out FCD, SNN, Scaf, Atomic/Molecular Stability in the main text. The legend inconsistency for FCD and the missing Novelty column in Table 2 will be corrected.
>
> ---
>
> ### W2: Performance
>
> We agree that, in the current configuration, NSGGM shows a clear trade-off. It achieves perfect validity and strong novelty but weaker distributional metrics on large molecular benchmarks.
>
> We want to provide two clarifications and planned improvements:
>
> 1. **Explicit trade-off and coverage.**
>    The model is intentionally configured with relatively coarse primitives (larger motifs, higher minimum support) to keep the vocabulary compact and solving fast; this narrows the effective support of the model and explains low scaffold diversity and elevated FCD. In a revision, we will:
>  Report **vocabulary sizes** and **coverage statistics** (fraction of training molecules exactly representable by the current token set) for all datasets.
>  Add ablations varying the minimum support and motif size bounds, showing how vocabulary size, scaffold diversity, FCD, and KL co-vary. This will make the trade-offs explicit rather than only discussed qualitatively.
>
> 2. **Potential performance improvements.**
>    We view NSGGM as a framework rather than a fixed model. Its distributional fit can be improved by:
>    - Using **finer-grained motifs** (smaller, more granular subgraphs) to expand the overall expressivity.
>    - Incorporating richer **blueprint priors** (like learned global constraint heads on top of the decoder, or simple reweighting schemes) while preserving the symbolic layer.
>
>
>    We will include preliminary results or discussion along these lines to clarify that the current numbers reflect a particular design point, not a fundamental limitation of the approach.
>
> ---
>
> ### W3: Evaluation of constraint satisfaction / controllability
>
> We agree that the current paper emphasizes qualitative examples (Figure 2) and that a **quantitative** evaluation of controllability would strengthen the case.
>
> In a revision, we plan to add experiments that:
>
> - Define a set of **non-trivial structural constraints** (required scaffold, ring-count bounds, distance constraints, forbidden structures) and measure:
>   - Constraint satisfaction rate,
>   - Validity/Novelty, and
>   - FCD / scaffold diversity under these constraints.
> - Compare NSGGM to **conditional generation baselines** where constraints are implemented via conditioning or post hoc filtering.
>
> This will allow us to quantify the core advantage: NSGGM can enforce new constraints *post-training* via the symbolic layer, whereas other baselines often require finetuning to approximate satisfaction.
>
> ---
>
> ### Usefulness of guaranteeing validity
>
> We agree that in some domain settings, “high but not perfect validity + rejection” is acceptable. Our goal is different: NSGGM targets applications where:
>
> - Constraints are **complex and task-specific**, and
> - Users want interpretable **guarantees** that any accepted sample satisfies them, without having to rely on black-box behavior or extensive rejection sampling.
> ---
>
> ### Relation to fragment-based / GA methods
>
> We appreciate you for pointing out fragment-based and genetic approaches. Conceptually, NSGGM is close at a high level: it builds molecules from reusable fragments. The key differences are:
>
> - The fragment inventory and assembly rules are **learned from data** via our decomposition + mining pipeline rather than hand-crafted.
> - The assembly phase is phrased as a **formal SMT problem** with correctness guarantees and a clean interface for additional logic-level constraints.
>
> We will add a dedicated discussion comparing NSGGM to fragment-based and GA frameworks and clarify how our learned primitives and symbolic assembly relate to these traditions.
>
> ---
>
> ### Additional remarks
>
> - **Large ring structures in Figure 6.**
>   These arise because the current constraints do not penalize or bound certain ring sizes. This is an example where adding explicit constraints (limiting ring size or requiring more realistic ring patterns) would prevent such unrealistic structures. This is precisely the kind of application we aim to support. We will add this discussion and an example with such constraints enabled.

---

> > ### Comment · Reviewer_DRHy · 2025-11-25
> >
> > I thank the authors for their thoughtful replies to my comments.
> > Incorporating all the improvements mentioned by the authors will indeed strengthen the contribution of this work in a future submission.
> > I agree with the authors that SOTA performance is not the be-all and end-all when introducing a new approach, and that finding another point on the pareto frontier by applying a "trade-off" can carry value. However, the difference in performance to other tools here is too stark. In a subsequent version, I thus suggest that the authors implement their suggested performance improvements, and increase the rigor, with which their results are presented.

---

### Official Review · Reviewer_LHQ7 · 2025-10-31

**Soundness:** 2
**Presentation:** 3
**Contribution:** 2
**Rating:** 2
**Confidence:** 4

**Summary:**

The paper proposes a novel generative model for molecules using a symbolic approach. Specifically, it encodes user/application constraints explicitly and uses an SMT solver (in combination with an autoregressive decoder generating samples) to check them.

**Strengths:**

- It's an interesting idea to use a logical representation and tools in the molecule representation setting.

**Weaknesses:**

- The questioning of diffusion models seems somewhat wrong-placed given the recent rather impressive results [1].
- To me, it is unclear if SMT solvers are really needed/useful here. Even the older, existing methods achieve rather high validity rates. Moreover the solver is only applied, for checking, so the logic does not seem to be encoded within the model.
- Apart from UniGEM, which is dropped after Table 1 (why?), the considered baselines are all rather outdated (2019-2023). There are various more recent ones, e.g., [2-4].
- Because of the former, it is impossible to judge the potential impact and advantages of the approach.

[1] https://hannes-stark.com/assets/boltzgen.pdf

[2] Ketata et al. Lift your molecules: Molecular graph generation in latent euclidean space. ICLR'25

[3] Wang et al. Learning-Order Autoregressive Models with Application to Molecular Graph Generation. ICML'25.

[4] Lee et al. GenMol: A Drug Discovery Generalist with Discrete Diffusion, ICML'25

**Questions:**

--------------------------------

---

> ### Author Response · Authors · 2025-11-22
>
> We thank the reviewer for the insightful feedback and for pointing us to additional recent work.
>
> **W1 / W3: Baselines and recency ([1–4])**
> We agree that including more recent diffusion and autoregressive models is important to assess impact. In a future revision, we will extend our experimental section to include recent methods listed and update our discussion to reflect their strengths and differences relative to our work.
>
> **W2: Usefulness of the SMT / logic layer**
> We agree that high validity on standard benchmarks can already be achieved by many neural models. Our goal is different: NSGGM is designed for **controllability** and **interpretability** rather than validity alone. The autoregressive decoder learns a data-driven proposal distribution, while the SMT layer enforces user-specified structural constraints **post-training**, at generation time. This lets users add or modify a broad class of constraints (i.e, required/forbidden substructures, ring-count bounds, connectivity patterns) without retraining or fine-tuning, and with exact enforcement whenever the constraints are feasible. On the other hand, most existing methods are predominantly neural, with constraints implemented implicitly through training or heuristic filtering, and do not expose an explicit, editable symbolic constraint layer.
>
> **W4: “Because of the former, it is impossible to judge impact”**
> This essentially reiterates the concern about the recency and breadth of baselines. We agree that more up-to-date comparisons will make the impact easier to judge, and we will address this by adding newer baselines and clarifying the positioning of NSGGM as a **controllable, interpretable neuro-symbolic alternative** that complements recent SOTA models rather than aiming to replace them.

---

> ### Comment · Reviewer_LHQ7 · 2025-11-23
>
> I'm confirming that I read the authors' response and also the other reviews and rebuttals. I think my evaluation is in line with the ones of the other reviewers.

---

### Official Review · Reviewer_qbzv · 2025-11-01

**Soundness:** 3
**Presentation:** 2
**Contribution:** 2
**Rating:** 4
**Confidence:** 4

**Summary:**

The authors introduce Neuro-Symbolic Graph Generative Modeling (NSGGM), a novel neural-symbolic framework reframing graph generation as a dual problem of sequence generation and constraint-satisfaction problem solving. An autoregressive decoder proposes a blueprint sequence of subgraph tokens, which a symbolic SMT solver then assembles into a final graph, providing the advantages in controllability and verifiability. Extensive experiments shows that the superiority of NSGGM in metrics Validity and Stability.

**Strengths:**

- The authors propose a novel neural-symbolic framework to treat graph generation as a compositional task.
- The proposed neural component, a lightweight autoregressive decoder, achieves faster training compared to diffusion-based approaches.
- The experimental evaluation is comprehensive, benchmarking against several state-of-the-art graph generation models across multiple graph-related datasets.

**Weaknesses:**

- While NSGGM ensures validity and stability in molecular generation tasks, it struggles to encode certain molecular constraints (e.g., uniqueness or stereochemical rules) that SMT solvers cannot easily capture. This limits its applicability to novel drug design, where such constraints are critical.
- The generative capacity of NSGGM appears limited by the fixed vocabulary size, which constrains the diversity and novelty of generated graphs. As reflected in the results, diversity and novelty metrics degrade substantially, even though validity and stability remain high.

**Questions:**

- How do the authors determine the granularity of subgraph tokens, *i.e.*, the smallest blueprint or atomic unit used in graph decomposition?
- Could the authors elaborate on the meaning of hard and soft constraints? It is unclear why these constraints are categorized as structural constraints.
- The paper would benefit from an analysis of how vocabulary size and subgraph decomposition affect the trade-off between validity, diversity, and novelty in the generated graphs.
- What is the inference latency of NSGGM compared to diffusion-based methods such as DiGress?

---

> ### Author Response · Authors · 2025-11-22
>
> We thank the reviewer for the thoughtful and detailed comments. We address the main concerns and questions below.
>
> **W1: Constraints (uniqueness / stereochemistry)**
> Our work currently focuses on 2D molecular graphs with standard atom/bond attributes and valence-based constraints. In this method, SMT constraints already enforce a rich set of structural and chemical hard constraints: subgraph integrity, edge-slot matching, element consistency, and valence satisfaction. On top of this, custom user constraints $\phi_{user}$ allows for additional structural and chemical-aware constraints like ring-count bounds, forbidden substructures, and element/functional-group constraints (Sec 3.2).
> - **Uniqueness.** This metric is defined on the sample-level. Enforcing “no duplicates across an entire sample set” is a global property and is not typically handled symbolically by generative models; the standard practice is post hoc deduplication, which we follow. This is a design choice, not a fundamental limitation of SMT.
> - **Stereochemistry.** We deliberately do not model stereochemistry in this work. Extending NSGGM to stereochemical labels is achievable in our formulation. Full 3D geometry would require coupling to geometric reasoning, which we view as future work rather than an inherent limitation.
> ---
> **W2 + Q3: Vocabulary, diversity, and novelty**
> We agree the trade-off between vocabulary size and diversity/novelty is important. In NSGGM, vocabulary granularity is a hyperparameter, not fixed: tree motifs are obtained by bounded-depth/size subtree mining with a minimum-support threshold.
> There are two observations:
> - Smaller motifs yield more combinatorial freedom and potentially higher diversity/novelty, at the cost of longer sequences and larger SMT instances.
> - Larger motifs do the opposite: more compact blueprints and faster solving, but a tighter distribution.
> In the current experiments we chose intermediate settings that give a compact vocabulary and fast training while achieving 100% validity and high novelty, with some loss in diversity metrics. In a revised version we will add an ablation varying motif size/support and report its effect on specific metrics.
> ---
> **Q1: How is token granularity chosen?**
> Granularity is determined by a deterministic decomposition pipeline:
> - Cycles from a minimum cycle basis are taken as cycle primitives.
> - Acyclic components are split via block–cut decomposition; within each block, bounded-depth/size subtrees are enumerated and frequent ones are mined as tree motifs.
> - Remaining fragments are covered by minimal trees / single edges.
> The subtree depth/size bound and minimum support are tuned on validation data and control the trade-off described above.
> ---
> **Q2: Hard vs. soft “structural” constraints**
> - **Hard constraints** enforce graph validity and consistency and must always be satisfied (subgraph integrity, type equality when merging copies, edge–slot matching, element and valence constraints for molecules, etc.).
> - **Soft constraints** define a penalty minimized subject to the hard constraints (e.g., matching target neighbor/slot counts and fused-cycle patterns).
> We refer to both as “structural” because they depend only on the graph’s connectivity and local slot/type annotations.
> ---
> **Q4: Inference latency vs. diffusion**
> Per sample, NSGGM performs: (i) a single autoregressive pass over the blueprint and (ii) one SMT solve, which dominates runtime for QM9-scale graphs. Diffusion models such as DiGress incur many denoising steps with cheaper per-step operations and are typically run on GPUs for reasonable throughput. In contrast, our SMT layer is inherently CPU-based and the autoregressive decoder can be kept small enough to run efficiently on CPU only. We report training-time comparisons in the paper. In a revised version, we will also include wall-clock sampling times and highlight this “CPU-friendly” benefit as a practical advantage of NSGGM.

---

> > ### Comment · Reviewer_qbzv · 2025-11-24
> >
> > I thank the authors for their response. After considering the rebuttal of authors and reading the comment of other reviewers, I believe the paper would benefit from major revisions in the following areas: (1) adding an ablation study on vocabulary size and (2) providing comparisons with state-of-the-art baselines. Therefore, I will keep my final score.

---

### Meta-Review · Area_Chair_nYnT · 2026-01-06

**Summary:**

The paper studies graph generation, casting it as a sequencing modeling with constraint satisfaction. The proposed model first learns a vocabulary of subraph tokens and then generates the subgraph sequences autoregressively.  Experimental results on molecular and generate graph benchmarks show the competitiveness of the proposed approach.

The paper studies an important problem. However, there are a few big limitations raised by the reviewers:

(1) the compared baselines are very old. It is difficult to justify the competitiveness of the proposed approach, especially considering the impressive results of recent diffusion based methods.

(2) experimental results are pretty weak, especially on metrics other than validity.


Due to the above limitations, the paper is not ready for publication at ICLR.

**Reviewer Concerns:**

In general, the rebuttal is not able to address all the reviewers' concern.

**Reviewer Scores:**

Due to the big limitations of the paper, it is difficult to convince the reviewers to change their scores.

---

### Decision · Program_Chairs · 2026-01-26

Reject